# Radar Target Radar Cross-Section Measurement Based on Enhanced Imaging and Scattering Center Extraction

**DOI:** 10.3390/s24196315

**Published:** 2024-09-29

**Authors:** Xin Tan, Chaoqi Wang, Yang Fang, Bai Wu, Dongyan Zhao, Jiansheng Hu

**Affiliations:** 1School of Electronic Information and Artificial Intelligence, Shaanxi University of Science and Technology, Xi’an 710016, China; tanxin@sust.edu.cn (X.T.); 221612141@sust.edu.cn (C.W.); 221612128@sust.edu.cn (B.W.); 221611048@sust.edu.cn (D.Z.); 2State Key Laboratory for Strength and Vibration of Mechanical Structures, Shaanxi Engineering Research Centre of NDT and Structural Integrity Evaluation, School of Aerospace Engineering, Xi’an Jiaotong University, Xi’an 710049, China; 3School of Information Engineering, Engineering University of PAP, Xi’an 710047, China; hujiansheng121@163.com

**Keywords:** RCS, enhanced imaging, scattering center extraction

## Abstract

Accurate measurement of a Radar Cross-Section (RCS) is a critical technical challenge in assessing the stealth performance and scattering characteristics of radar targets. Traditional RCS measurement methods are limited by high costs, sensitivity to environmental conditions, and difficulties in distinguishing local scattering features of targets. To address these challenges, this paper proposes a novel RCS measurement method based on enhanced imaging and scattering center extraction. This method integrates sub-aperture imaging with image fusion techniques to improve imaging quality and enhance the detail of target scattering characteristics. Additionally, an improved sequence CLEAN algorithm is employed to effectively suppress sidelobe effects and ensure the accuracy of scattering center extraction. Experimental results demonstrate that this method achieves higher precision in RCS measurement of complex targets and is particularly effective in environments with strong interference, where it successfully separates the scattering contributions of the target from those of the interference sources. This method offers a new technological approach for precise RCS measurement of radar stealth targets in the future.

## 1. Introduction

A Radar Cross-Section (RCS) is a crucial physical parameter for describing the scattering characteristics of radar targets [1]. With the rapid advancement in stealth technology and radar detection systems, the demand for precise RCS measurement has become increasingly urgent [2]. Traditional far-field and compact range RCS measurement methods, while playing a significant role in practical applications, face challenges such as high costs, susceptibility to environmental interference, and limitations related to target size. Moreover, these methods typically measure only the overall RCS of a target and are unable to provide detailed insights into the scattering characteristics of local structures [3]. This limitation restricts their application in the design, testing, and performance evaluation of stealth targets.

With advancements in high-resolution radar imaging technology [4], imaging-based RCS measurement methods have increasingly garnered attention. The core of radar imaging technology lies in its ability to capture multiple scattering centers of a target and reveal the spatial distribution and corresponding scattering intensity of these centers through imaging algorithms [5]. These techniques not only provide high-resolution imaging data under near-field conditions but also enable the extraction of spatial location [6], size, and shape information of the target through imaging analyses. Such methods can obtain both the overall scattering characteristics of the target and diagnose local scattering features using high-resolution imaging techniques. Furthermore, imaging-based RCS measurement methods have gained widespread attention due to their interference resistance capabilities and have led to several research developments in this area. Most of these methods use the BP algorithm for imaging, as evidenced by the studies in references [7,8,9], which utilize BP imaging algorithms for RCS measurement.

In 1998, J. Palau et al. discovered a transformation relationship between ISAR images and RCS [10], allowing the inversion of the target’s 2D RCS response from imaging. However, this method requires interpolating polar coordinates to Cartesian coordinates, introducing interpolation errors that reduce RCS measurement accuracy. In 2003, R.J. Burkholder proposed an inversion method based on time-domain imaging algorithms that avoids interpolation calculations, thereby improving measurement accuracy [11]. Sensani et al. introduced a near-field to far-field (NF2FF) processing program for radar images, which enhances the understanding of RCS scattering mechanisms [12]. In 2019, Alvarez J employed the CLEAN algorithm in near-field ISAR imaging to suppress sidelobes and mitigate ground reflections [13], but this approach was not specifically applied to the scattering inversion of complex targets. Moreover, while the method effectively suppresses sidelobes, it does not distinguish whether the scattering points are true scattering responses from the target.

When using imaging results to invert the RCS of a target, the quality of the imaging directly impacts the measurement accuracy of the final target RCS [14]. Due to the difficulty in achieving full sampling in both the frequency and azimuth for ISAR imaging, sidelobe clutter issues inevitably arise from data truncation. These sidelobe effects can significantly affect the accuracy of the inversion process, as the quality of the imaging directly influences the inversion precision. Traditional imaging algorithms often exhibit high sidelobes, which can lead to mutual interference among sidelobes. To mitigate the effects of sidelobe clutter, windowing methods are commonly employed; however, this approach increases the main lobe width and does not improve resolution [15,16,17]. In other words, traditional imaging methods suffer from sidelobe effects and noise interference, which can result in an inaccurate extraction of target scattering characteristics.

To address the aforementioned issues, this paper proposes a novel RCS measurement method based on enhanced imaging and the sequence CLEAN algorithm. This method combines sub-aperture imaging with image fusion techniques to improve imaging resolution. Additionally, the improved CLEAN algorithm is employed to suppress false scattering centers, significantly enhancing the accuracy of RCS measurements.

## 2. Enhanced Imaging and Scattering Center Extraction Techniques

### 2.1. Conventional Turntable Imaging

ISAR (Inverse Synthetic Aperture Radar) imaging technology utilizes a wide frequency band and large observation angles to generate high-resolution two-dimensional images, effectively characterizing the target’s scattering distribution [18]. However, conventional imaging methods often experience significant sidelobe effects in practical applications, leading to interference between signals from different scattering points [19,20]. This interference ultimately impacts the overall quality and accuracy of the imaging.

Figure 1 illustrates the turntable imaging model, where both the transmitting and receiving antennas are positioned at fixed locations relative to the target center. The target is placed on a turntable and rotated at fixed angular intervals. At each rotation angle, the antenna transmits a wideband signal and receives the echoed signals at the same position. The measured data result in a two-dimensional echo dataset of the target as a function of frequency and observation angle.

In near-field measurements, assuming there are N equivalent scattering centers, and the position of any one scattering point is $\left(x_{i},y_{i}\right)$, the echo signal received by the receiving antenna is given by
(1)$$E_{n}(f,\theta )=\sum_{i=1}^{N}\gamma (x_{i},y_{i})\frac{e^{-j(4\pi f/c)}}{R_{i}^{2}}$$
where $\gamma (x_{i},y_{i})$ represents the spatial reflectivity distribution of the target.

The filtered back-projection imaging algorithm is used to image the scattering echoes, which involves coherently summing the scattering echo signals at each imaging point.
(2)$$\begin{array}{ll} \gamma (x,y) & =\int_{\theta_{\min}}^{\theta_{\max}}\int_{k_{\min}}^{k_{\max}}E_{n}(k,\theta )\cdot R_{i}^{2}e^{j2\pi k_{\min}l}dkd\theta \\ & =\int_{\theta_{\min}}^{\theta_{\max}}\int_{0}^{B}E_{n}(k+k_{\min},\theta )\cdot e^{j2\pi kl}e^{j2\pi k_{\min}l}dkd\theta \end{array}$$
where $\theta_{\min}$ and $\theta_{\max}$ are the start and end angles of the observation angle, respectively; $B=k_{\max}-k_{\min}$ is the bandwidth of the spatial frequency k=2f/c; $k_{\min}$ is the spatial frequency of the start frequency; and l=ycosθ−xsinθ.

The azimuth and range resolutions of the ISAR imaging method are given by
(3)$$\delta_{x}=\frac{\lambda }{2\theta },\delta_{y}=\frac{c}{2B}$$

To visually analyze the effects of conventional imaging, the FEKO2020 software was used to model and calculate scattering data. Imaging was performed using the conventional back-projection (BP) method. Figure 2 shows the model of five metal spheres, each with a diameter of 0.06 m, arranged laterally. The distance between the centers of two adjacent spheres is 0.12 m. Figure 3 presents the imaging results.

Analyzing the imaging results of the five spheres, strong sidelobes are present for each scattering target, and sidelobe clutter exists between the targets. Although the scattering points in the two-dimensional image correspond to the simulation model, it is necessary to account for sidelobe effects and evaluate the robustness of this method to avoid false points caused by sidelobe interference when imaging complex targets. To address this, several ideal point targets were simulated. The true values of the simulated point targets are shown in Figure 4a, with the imaging range set from −0.8 m to 0.8 m, and the number of pixels is 471. The conventional BP imaging algorithm is utilized to image the target while taking noise into account and making the signal-to-noise ratio of SNR = 20 dB, which contains a total of 10 point targets in five groups. From left to right, the spacing between each pair of points in each group is 20 mm, 24 mm, 28 mm, 35 mm, and 40 mm, respectively. The simulation uses a frequency bandwidth of 4 GHz and an azimuth angle range of 30°. It can be observed that the azimuth resolution $\delta_{x}=0.0286\;m$, and the spacing between the first two groups is smaller than the system’s theoretical resolution.

The imaging results without noise and with noise interference are shown in Figure 4b and Figure 4c, respectively. From the imaging profile results, it can be observed that the conventional imaging method produces similar outcomes regardless of the presence of noise. In both cases, it fails to distinguish targets with spacings smaller than the system’s theoretical resolution. Furthermore, the sidelobe levels are relatively high. As shown in Figure 4c, the sidelobes of one target interfere and overlap with those of other targets. When the target’s scattering is weak, the target may be submerged in noise.

The main source of the sidelobe effect is the truncation of finite-length signals [21]. In the range direction, the limited signal bandwidth causes spectral leakage, while in the azimuth direction, variations in the observation angle leads to spectral leakage from the equivalent synthetic azimuth bandwidth. This truncation effect not only reduces image resolution but also results in the appearance of false scattering centers [22,23]. Therefore, reducing the sidelobe effect is crucial for improving imaging quality and the accuracy of RCS measurement.

### 2.2. Enhanced Imaging Using Sub-Aperture Synthesis and Image Fusion

Turntable imaging obtains high-resolution two-dimensional images of the target by utilizing a wide frequency band and large azimuth observation angles. This approach provides a scattering intensity distribution map of the target by coherently summing wideband and wide-angle sampled signals at each imaging point. The wide frequency band and large angle determine the range and azimuth resolutions of the image. To capture more scattering information about the target, data from a wide azimuth angle range are combined to improve the azimuth resolution. However, considering the anisotropic scattering characteristics of the target, the equivalent scattering centers of the target will vary with different viewing angles [24]. To visually analyze the target’s anisotropic scattering, electromagnetic data from a CAD model of an aircraft were simulated. The CAD model of the aircraft is shown in Figure 5. The experimental simulation parameters are as follows: a center frequency of 10 GHz, a bandwidth of 4 GHz, sampling interval is 40 MHz, an azimuth angle range from −30° to 30°, and an angle interval of 0.2°. Figure 6a–d shows the target images generated using different sub-aperture echo data, with sub-aperture center angles of −12°, −2°, 2°, and 10°, respectively.

Since the aircraft model is symmetric about the *x*-axis, the imaging results in Figure 6b,c exhibit mutual symmetry. It is also evident that the target scattering characteristics vary significantly in the images generated with different sub-apertures, indicating that radar targets exhibit anisotropy under large-angle observations. For a fixed imaging sub-aperture center, the scattering characteristics of the target from other observation angles are inevitably lost.

For this reason, this paper proposes a method based on sub-aperture synthesis and image fusion to enhance imaging quality. First, the entire large observation angle is divided into several small-angle sub-apertures, and imaging is performed for each sub-aperture individually. Then, all sub-images are registered through rotation alignment, and finally, a combined image is generated through incoherent summation.

We assume that the full aperture is uniformly divided into L sub-apertures, where the echo data of the i-th sub-aperture can be expressed as
(4)$$E_{i}(f,\theta )=E(f,\theta )rect\left[\frac{\theta -\Delta /2-(i-1)\theta_{c}}{\Delta }\right]$$
where rect(⋅) denotes the rectangular window function, Δ represents the azimuth accumulation angle of the sub-apertures, and $\theta_{c}$ is the angular difference between adjacent sub-aperture centers. To prevent significant discontinuities in the accumulation angles between sub-apertures, we protect weak scattering points with short azimuth responses, and improve spectral utilization, and the general rule is to make $\Delta >\theta_{c}$, there is a certain overlap angle between neighboring sub-apertures, and the shortening of the length of sub-apertures reduces the influence of anisotropy on imaging.

However, in real scenarios, it is not possible to determine the precise division criteria for the full-aperture azimuth sub-apertures based on prior information about the target. Therefore, an adaptive division mechanism is required. Based on the initial division of the echo data in Equation (4), each uniform sub-aperture is first imaged, and the imaging result of the sub-aperture divided by the center angle is set to be $\gamma_{0}(x,y)$, and the imaging result of its adjacent sub-apertures is set to be $\gamma_{1}(x,y)$, where a correlation coefficient is defined as follows:(5)$$R=\frac{sum(sum(\gamma_{0}(x,y)⊗\gamma_{1}(x,y)))}{\sqrt{{\left\|\gamma_{0}(x,y)\right\|}_{2}*{\left\|\gamma_{1}(x,y)\right\|}_{2}}}$$

R denotes the degree of variation between the imaging results of the two datasets. ⊗ denotes Hadamard multiplication, and ${\left\|∙\right\|}_{2}$ indicates the $L_{2}$ norm.

If the inter-correlation coefficient between $\gamma_{0}(x,y)$ and $\gamma_{1}(x,y)$ is large, the data range corresponding to $\gamma_{1}(x,y)$ is adapted to expand or reduce to meet the threshold to delineate the first sub-aperture angle adjacent to the center angle. Subsequently, the center angle of the first determined sub-aperture angle is used as a reference to divide the remaining sub-aperture angles.

The sub-aperture images only represent the target scattering distribution within their corresponding observation angles. To obtain the scattering information for the full aperture, all sub-images need to be rotated, aligned, and combined. A pixel-level fusion method is employed, which integrates multiple sub-images by considering their original data. Based on spatial domain linear weighting fusion criteria, each pixel of the sub-images is processed with a weight to obtain the enhanced image pixels. The fusion of sub-images is performed according to Equation (6).
(6)$$\gamma (i,j)=w_{1}\gamma_{1}(i,j)+w_{2}\gamma_{2}(i,j)+\cdots +w_{n}\gamma_{n}(i,j)$$
where (i,j) represents the pixel location in the image; $w_{1},w_{2},\ldots w_{n}$ denotes the weights; and the weight is determined by considering the contribution of each pixel in the sub-images to the average imaging. Specifically, the weight is set as 1/N.

Imaging of the model is in Figure 5, using the proposed method with an aperture size ranging from −30° to 30° and a frequency range of 8 to 12 GHz. The sub-aperture synthesis was performed using the proposed method, and the imaging result is shown in Figure 7.

In this process, the adaptive division mechanism plays a crucial role. By calculating the inter-correlation coefficients of adjacent sub-aperture imaging results, the division of sub-apertures is dynamically adjusted to ensure the coherence of the imaging results. The final combined image more accurately reflects the true scattering distribution of the target. Figure 7 shows the enhanced imaging results obtained using the proposed method. Compared to conventional imaging, the enhanced imaging method provides a clearer depiction of the target’s true scattering distribution. Although sidelobe effects still persist, subsequent scattering center extraction techniques can further suppress these sidelobes.

### 2.3. RCS Inversion Based on Sequence CLEAN Technique

In practice, the two-dimensional imaging result of the target represents the distribution of the target’s reflectivity. To extract the RCS of the area of interest, we can directly use a two-dimensional window function to segment the region within the image. This results in a new imaging map:(7)$$\gamma_{0}(x,y)=\left\{\begin{array}{ll} \gamma (x,y) & r<a\\ 0 & r\geq a \end{array}\right.$$

The new two-dimensional image is transformed into the spectral domain using a two-dimensional Fourier transform:(8)$$E(K_{x},K_{y})=\int_{y_{1}}^{y_{2}}\int_{x_{1}}^{x_{2}}\gamma_{0}(x,y)e^{j2\pi (K_{x}+K_{y})}dxdy$$

The interpolation of $E(K_{x,}K_{y})$ yields E(f,θ) that varies with frequency and angle, and the interpolation equation is
(9)$$f=(c/2)\times \sqrt{K_{x}^{2}+K_{y}^{2}}$$
(10)$$\theta =-\tan^{-1}(K_{x}/K_{y})$$

Similarly, for a calibration target, RCS is measured using the described imaging and extraction methods, resulting in spectral domain data $E_{0}(f,\theta )$. By comparing the target’s spectral domain data with that of a metal sphere, the RCS of the measured target is obtained:(11)$$\sigma =E(f,\theta )-E_{0}(f,\theta )+\sigma_{0}$$
where $\sigma_{0}$ is the theoretical RCS of the calibrator.

RCS inversion is performed on the two types of imaging results and compared with the MLFMM calculation results, as shown in Figure 8.

The accuracy of RCS inversion using the enhanced imaging method has significantly improved compared to conventional direct inversion methods. However, an error of up to 6 dB still exists within ±3 to ±4 degrees of the azimuth angle. To update and improve the above imaging method, the effective application of the joint sequence CLEAN technique in eliminating the sidelobes, the image is re-synthesized.

While the enhanced imaging method accounts for anisotropic scattering, it does not effectively suppress sidelobes for each scattering point nor distinguish whether the points on the image are genuine scattering responses. The scattering distribution may be affected by the overlap of sidelobes from closely spaced scattering points or the interaction between main lobes and sidelobes. This can result in a scattering distribution in the image that does not accurately represent the true target points, with strong scattering points potentially causing false targets and displacement of the main lobe.

A new CLEAN technique is employed to re-extract the scattering points [25,26]. The basic idea of the algorithm is as follows: even if the point with the highest amplitude in the image is a false target, it may still represent a genuine scattering response with low intensity. By removing both the main lobe and sidelobes of this point, the signal energy is reduced, exposing weak targets that are obscured by strong scattering interference. If each target reduced during the CLEAN iteration is a true scattering point, the remaining signal energy will be minimized after the CLEAN process.

The steps of the algorithm are as follows:The kth iteration selects the n peaks in the image
$\gamma_{0}$ and calculates the signal energy:
(12)$$E(k)=\int_{-\infty }^{+\infty }\gamma_{0}(x,y)\gamma_{0}^{*}(x,y)dxdy$$
The dirty point spread function $\gamma_{0,i}(x,y)=\gamma_{0}(x,y)-A\cdot e^{j\varphi }\cdot PSF(x-x_{i},y-y_{i})$ is subtracted from the image for the first (i=1,2,⋯n)
peak record searched, corresponding to its complex value $A\cdot e^{j\varphi }$
and position $\left(x_{i},y_{i}\right)$.
Calculate the energy of the new signal $\gamma_{0,i}(x,y)$, $E_{i}(k+1)=\int_{-\infty }^{+\infty }\gamma_{0,i}(x,y)\gamma_{0,i}^{*}(x,y)dxdy$.Compare the energy of the new signal with that of the signal before the reduction, and if $E_{i}(k+1)<E(k)$, then this reduction is considered to be a real target; record the position and amplitude of this point, set k = k + 1, and repeat steps 1–3.If $E_{i}(k+1)>E(k)$, the current reduction is considered to be a false target and the iterative process is terminated.Repeat steps 2–5 for n peaks.If the energy of the signal after the final iteration converges, terminate the entire iterative process.A clean image is obtained by convolving the ideal point spread function with the recorded points.


A new two-dimensional image is generated by applying the scattering center extraction method to the image shown in Figure 7, as presented in Figure 9.

The results of the inversion are plotted against the RCS calculated by MLFMM in Figure 10.

From Figure 10, it is evident that the RCS inversion results at various azimuthal angles closely match the computed results. However, there is a noticeable discrepancy at the ends of the azimuthal angle range, which is attributed to data truncation effects. Overall, the accuracy of RCS measurements within the azimuthal angle range has been significantly improved.

### 2.4. Complete Workflow of Proposed Method

The overall workflow for the imaging inversion method is illustrated in Figure 11. The process involves enhancing the imaging for both the calibration target and the test object, followed by extracting the two-dimensional image of the area of interest. Subsequently, the sequence CLEAN method is applied to recompose a clean image. Finally, the RCS of the target is inversely derived using the transformed relationship between the image and the echo electric field.

## 3. Experimental Analyses

### 3.1. Simulation Analysis

#### 3.1.1. Simulation Experiment 1

The simulation parameters are as follows: a center frequency of 10 GHz, a bandwidth of 4 GHz, the sampling interval is 40 MHz, and an azimuth angle range from −15° to 15° with 151 sampling points. The model consists of a combination of five metal spheres, each with a diameter of 60 mm and a center-to-center spacing of 120 mm between adjacent spheres.

The model shown in Figure 12 is used to validate the proposed enhanced imaging method and the combined sequence CLEAN technique. The scatter point distribution formed by the equivalent scattering centers of the synthesized target, obtained through imaging extraction, is illustrated in Figure 13.

Compared to the conventional imaging method shown in Figure 3, it is evident from the two-dimensional image that the sidelobe crosstalk between targets has been completely eliminated, thereby mitigating the direct impact of sidelobe levels on subsequent RCS inversion.

To analyze the accuracy of scatter center extraction, the sequence CLEAN technique was applied to synthesize the target image. This process involves locating the positions of true scatter centers. Table 1 lists the relative position coordinates of the simulation model and the coordinates of the extracted scatter centers.

The range and azimuth resolutions are 0.0375 m and 0.0286 m, respectively, with an imaging cell size of 0.05 m, and the imaging scene is 0.71 m ∗ 0.71 m, based on the positional error; the maximum deviation in the range direction is 0.015 m; and the maximum deviation in the azimuth direction is 0.005 m. Compared to the resolutions of 0.0375 m and 0.0286 m in the two directions, the extraction positional error is within one imaging resolution cell.

For the azimuthal profile analysis of the target imaging, the Peak Sidelobe Ratio (PSLR) is assessed to evaluate the imaging quality for the simulated discrete single composite target. The azimuthal profiles of both the conventional imaging and the synthesized image after scatterer extraction are analyzed. Figure 14 shows the profile for conventional imaging, with a PSLR of −15.77 dB. In radar signal processing systems, a PSLR greater than 30 dB is typically required. Therefore, the conventional imaging result has significant sidelobes and does not meet the requirement. In contrast, the image obtained after scatterer extraction exhibits no sidelobes, with the PSLR corresponding to the maximum intensity of the scatterers.

We imaged the target model in Figure 12 using both the traditional method and the sequence CLEAN method. The scattering points from the central horizontal cross-section were extracted and normalized. The resulting two-dimensional horizontal cross-sectional images, shown in the Figure 15, illustrate the distribution of main lobes and sidelobes for both methods. It is evident that the proposed method significantly suppresses the sidelobes between scattering points. Compared to the traditional method, the main lobe remains unchanged in width, while the sidelobes decay much more rapidly.

The Integration Sidelobe Ratio (ISLR) refers to the ratio of the energy outside the impulse response resolution cell to the energy within the resolution cell. Equation (13) provides the formula for calculating the ISLR.
(13)$$ISLR=10*\log_{10}(I\_main/I\_side)$$

I_main represents the energy within the main lobe, while I_side represents the energy within the sidelobes.

A lower ISLR in SAR images indicates that the dark areas in the image are less likely to be affected by adjacent strong scatterers. This paper uses the Integration Sidelobe Ratio (ISLR) to quantitatively assess the advantages of the proposed method over traditional methods in reducing sidelobes. The quantitative results are presented in Table 2.

By extracting the equivalent scattering centers to synthesize the image and using this to back-calculate the RCS, the effectiveness of the proposed method in improving RCS inversion accuracy is validated. This is compared with conventional direct extraction, enhanced imaging extraction, proposed method extraction, and simulated theoretical data. The final RCS inversion results for various methods are shown in Figure 16.

In the inversion results shown in Figure 16, the comparison between the results obtained using the proposed method and those obtained through direct imaging extraction demonstrates that the proposed method not only effectively suppresses sidelobes but also significantly improves the RCS measurement results.

The comparison of conventional imaging with high sidelobes for direct RCS inversion, enhanced imaging using sequence CLEAN technology for extracting equivalent scattering centers, and simulated theoretical RCS using FEKO software reveals significant differences. Due to the sidelobe interference in conventional imaging, the inverted RCS significantly deviates from the theoretical RCS at most azimuth angles. In contrast, the proposed scattering center synthesis method shows a much higher correlation with the theoretical RCS, achieving improved inversion accuracy over direct imaging.

Although the proposed method effectively suppresses the influence of sidelobe interference between the target and the surrounding area, it does not yet confirm its generalization capabilities in the presence of strong interference sources around the target. To validate whether the proposed method can efficiently invert RCS under strong interference conditions, a dihedral structure was introduced behind the combined structure in Figure 12. The dihedral structure causes multiple scattering inside, leading to coupling effects with the combined structure’s scattering. This serves as interference to test the proposed method’s robustness in efficiently inverting RCS under a high-interference environment.

The imaging results of the model in Figure 17 are shown in Figure 18.

As shown in Figure 18a imaging results, the conventional imaging method can image both the interference source and the target region. However, there is a significant issue of sidelobe crosstalk between individual targets. Using the proposed enhanced imaging method, only the target scattering points are retained, and the sidelobe effects are largely filtered out.

The imaging result of the combined model of five spheres was extracted, with the interference source (dihedral) directly filtered from the imaging result. The extracted target’s two-dimensional data were then transformed into the spectral domain for RCS inversion. The comparison of the RCS results obtained from the proposed imaging method, the theoretically calculated RCS of the target region, and the RCS calculated with the dihedral interference present is shown in Figure 19.

The solid line represents the RCS theoretically calculated for the entire target, including the dihedral interference. The black dashed line corresponds to the RCS of the spherical array reconstructed through image extraction. Compared with the curve containing interference, notable fluctuations occur in the angular domain between individual spheres, especially at the frontal view. This is primarily due to mutual coupling between the targets containing the interference source that was not corrected through image extraction. By extracting the spherical array’s image from the interfered target and comparing it to the theoretical RCS of the target without interference, it is evident that the proposed imaging extraction method can effectively separate the scattering contributions of the interference source from those of the target.

In this process, the complete RCS reconstruction of the target’s overall structure and the separation of target and interference scattering under strong interference conditions were achieved. The proposed method demonstrates the capability to extract the local structure of the target and inversely compute its local scattering characteristics. The imaging and RCS inversion accuracy of the central sphere from the combined spherical structure is verified by comparing it to a simulated identical target.

The image of the region of interest was extracted from the overall 2D image shown in Figure 13. The extraction results are presented in Figure 20. Subsequently, the 2D image data were transformed into the spectral domain using a 2D Fourier transform for RCS inversion. The RCS results obtained from various imaging methods were compared with the MLFMM calculation results, as shown in Figure 21.

Using the MLFMM-calculated results as a reference, the error curves comparing the directly extracted inversion, the enhanced imaging with coherent CLEAN inversion, and the proposed method’s inversion results against the reference RCS are illustrated in Figure 22.

According to Figure 22, for simple geometric targets, the RCS error using both the direct extraction method and the proposed method ranges from −48 dBsm to −33 dBsm. Here, the RCS error is defined as the difference between the RCS obtained using the proposed method and the reference value, initially measured in m^2^, and then converted using $10*\log10(error(unit(m^{2})))$, with the result expressed in dBsm. However, the direct extraction method includes sidelobes within a certain range around the scattering points when extracting the image, leading to errors fluctuating between −48 dBsm and −33 dBsm compared to the MLFMM simulation results. The enhanced imaging combined with the sequence CLEAN technique effectively suppresses sidelobe interference from surrounding targets on the central sphere, keeping the inversion error consistently within approximately −39 dBsm across various azimuths. Overall, the proposed method significantly improves the accuracy of RCS measurement.

To investigate the angular aperture limit for accurate RCS inversion based on imaging results, we applied the model shown in Figure 17. Using a 0° angle as the aperture center, we performed RCS inversion within ±10° azimuth angles based on simulation data covering ranges of 30°, 24°, and 20°. The comparison results are illustrated in Figure 23.

The mean absolute error and root mean square error were used to judge the error at each size aperture.

Based on the data presented in Table 3, it can be observed that as the aperture size decreases, both the mean absolute error (MAE) and root mean square error (RMSE) increase, though the changes are relatively minor. This indicates that when calculating the target RCS using imaging methods, increasing the azimuth observation angle can enhance image resolution without significantly affecting the RCS accuracy for specific azimuth angles. However, the aperture size limit may vary depending on the model structure, necessitating a further analysis for specific tests.

It is well known that the azimuthal resolution is related to the size of the synthetic aperture and the center frequency. In the above experiment, using the model shown in Figure 16 with a distance of 0.12 m between the centers of two spheres, the azimuthal resolution in experiments with different aperture angles was always less than the distance between the targets, meaning that any two targets could always be resolved. Thus, the proposed method achieves high measurement accuracy for RCS when targets are resolvable. To verify the effectiveness of the proposed method when targets are not resolvable, we simulated the model in Figure 16 with the distance between the centers of the two spheres set to 0.08 m. With an azimuthal aperture angle of 10° and a test frequency band of 8–9 GHz, the azimuthal resolution calculates to 0.1011 m, and the range resolution calculates to 0.15 m. The comparison of the RCS from the imaging calculation model with the theoretical RCS is shown in Figure 24.

The results show that when the resolution exceeds the minimum resolvable distance between targets, the proposed method offers a better fit with the data curve compared to traditional methods. This indicates that, regardless of whether the resolution meets the resolvability criteria, the proposed method integrates all scattering information from the targets to the greatest extent possible, thereby providing higher measurement accuracy.

#### 3.1.2. Simulation Experiment 2

The specific simulation parameters are as follows: a center frequency of 10 GHz, and a bandwidth of 4 GHz, with 101 sampled frequency points. The azimuth angle range is −15° to 15°, with 151 sampling points. Figure 25 shows the simulation target, which is a sprayer pod. The total length of the target is 0.2 m.

The verification was carried out based on the given target model. The imaging results of both the traditional imaging method and the proposed method are shown in Figure 26a and Figure 26b, respectively.

From the visual analysis of the 2D imaging results, the traditional imaging method exhibits speckle noise in the range direction, whereas the proposed method effectively filters out clutter outside of the target, resulting in a cleaner background level.

To further analyze the sidelobes in the 2D images more intuitively, the central range profile was extracted for both imaging methods. The range profile of the traditional method and the scattering center extraction method’s central range profile are shown in Figure 27.

Figure 27 shows the normalized results. In the range profile obtained using the traditional method, the central point corresponds to the target area, and the sidelobes on both sides of the main lobe exhibit oscillations characteristic of an SINC function. These oscillations indicate that the 2D image generated by the traditional method is affected by significant clutter. However, analyzing the results of the proposed method in Figure 27, the central position corresponds to the target area, and the middle section represents the scattering points that correspond to the target’s physical structure. Outside the target area, the sidelobes in the profile are clearly visible. From the vertical axis of the graph, it is evident that the sidelobe values drop sharply compared to the traditional method, effectively suppressing clutter.

To quantitatively assess the sidelobe suppression effect, we also used the integrated sidelobe ratio (ISLR) as an evaluation metric. A lower ISLR indicates that dark areas in the image are less affected by nearby strong scatterers. The quantitative results are shown in Table 4.

The analysis above demonstrates the advantages of the proposed method in sidelobe suppression, which serves as the basis for comparing the final RCS measurement accuracy. The final RCS measurement accuracy is calculated using both the traditional method and the enhanced imaging scattering center method proposed in this study. The results are then compared with the theoretical RCS values, as shown in Figure 28.

The black dashed line and the blue solid line represent the RCS curves obtained from the traditional method and the proposed method, respectively. In the azimuth range of −15° to −5°, the RCS calculated by the traditional method shows a significant relative error compared to the theoretical value. This discrepancy primarily arises from the tip effect of the cone and the mutual coupling of various small components, which introduce clutter and false points in the two-dimensional image. During the scattering inversion process, these false points and clutter negatively impact the RCS calculation. In contrast, the curve generated by the proposed method aligns closely with the theoretical values, demonstrating high-quality measurement accuracy.

### 3.2. Experimental Validation

In the microwave anechoic chamber, an aircraft model was placed alongside a metal sphere with a diameter of 150 mm as the calibration target. Optical photos of the two targets are shown in Figure 29 and Figure 30. The test frequency range was 8 to 12 GHz, the sampling interval is 40 MHz. The rotation angle of −10°~10°, and the angle interval of 0.2 degrees. Figure 31a illustrates the imaging results with positional offsets for different aperture centers, while Figure 31b presents the distribution of the target’s scattering centers synthesized using the proposed method.

Complex targets exhibit continuous distribution and are affected by anisotropic scattering, causing the positions of scattering points to vary under different sub-apertures. A single coordinate in the image may represent different scattering points with varying intensities depending on the aperture size. Figure 31a visually demonstrates that the same scattering point can shift in imaging position and change in scattering amplitude under different azimuthal angles. By integrating large aperture data, the proposed method effectively extracts equivalent scattering centers to synthesize the target scattering distribution. This approach ensures consistency of scattering points across different sub-apertures, effectively suppressing sidelobe interference and spurious points, and enhancing the integrity of azimuth-dependent scattering information extraction. The extracted image’s scattering point distribution corresponds well with the actual target scattering points.

After transforming the imaging results to the spectral domain and applying inversion and calibration processes, the RCS variations in the target at different frequency points with respect to azimuthal angles are obtained. Figure 32 shows the RCS variation curves with azimuthal angle changes.

The proposed enhanced imaging method combined with sequence CLEAN for scatterer extraction proves feasible for RCS inversion within the azimuthal angle range. The inversion results are consistent with the actual measured RCS values. However, a significant error is observed at −3.4°, attributed to the coupling effects of the target’s structure on electromagnetic waves during observation and the coherent accumulation of adjacent scatterers during imaging and extraction.

Figure 33 shows the frequency response curve of the RCS with a test frequency range of 8–12 GHz. According to the experimental results, the RCS measurement curves obtained using the proposed method exhibit good agreement with the true RCS curves at various frequency points. This indicates that the method demonstrates practical applicability.

## 4. Conclusions

This paper presents a target RCS measurement method based on enhanced imaging and sequence CLEAN techniques. By combining sub-aperture imaging with image fusion techniques, the proposed method improves imaging quality and effectively suppresses sidelobe effects through an improved sequence CLEAN algorithm, significantly enhancing RCS measurement accuracy. Experimental validation demonstrates that the proposed method exhibits strong robustness and high measurement precision in complex target scenarios and strong interference environments, providing a novel solution for RCS measurement in practical engineering applications.

## Figures and Tables

**Figure 1 sensors-24-06315-f001:**
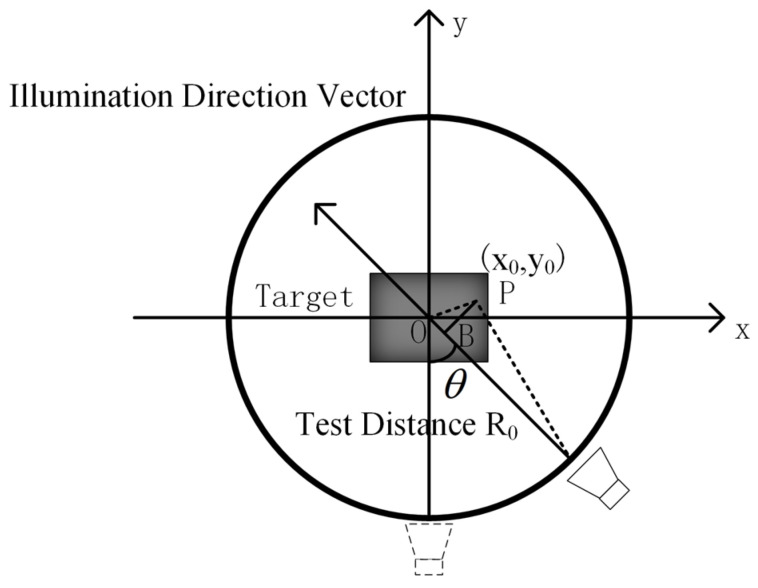
Imaging model of turntable.

**Figure 2 sensors-24-06315-f002:**
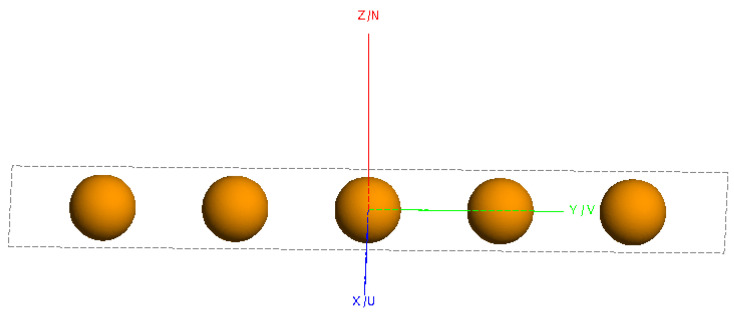
Simulation model of five metal spheres.

**Figure 3 sensors-24-06315-f003:**
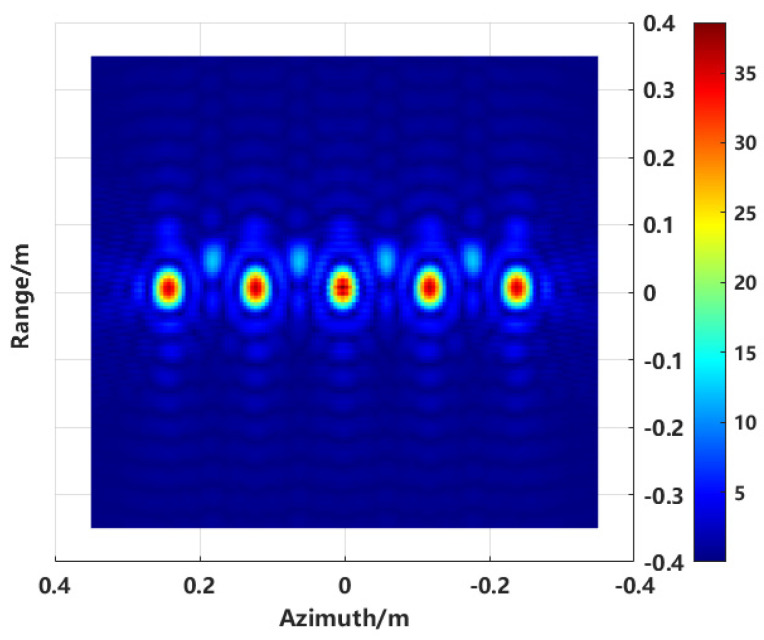
Conventional BP imaging result.

**Figure 4 sensors-24-06315-f004:**
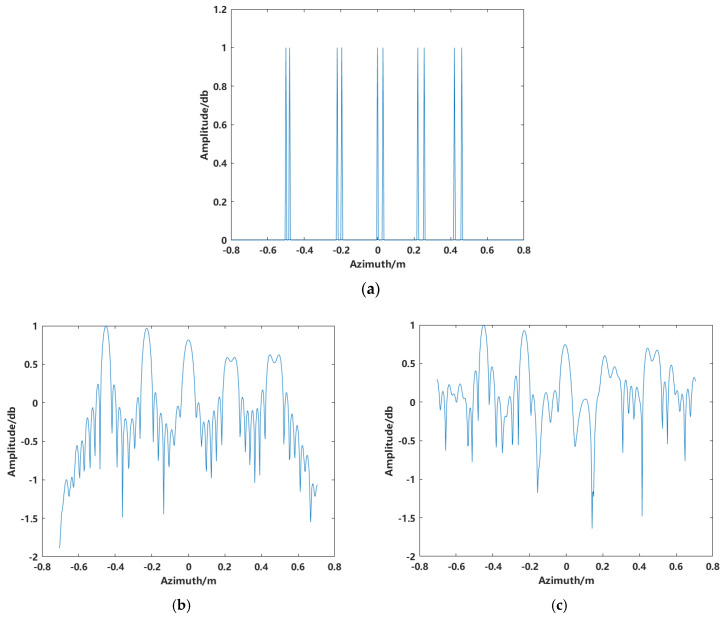
1D azimuth imaging results. (**a**) True values of point targets, (**b**) imaging without noise, and (**c**) imaging with noise.

**Figure 5 sensors-24-06315-f005:**
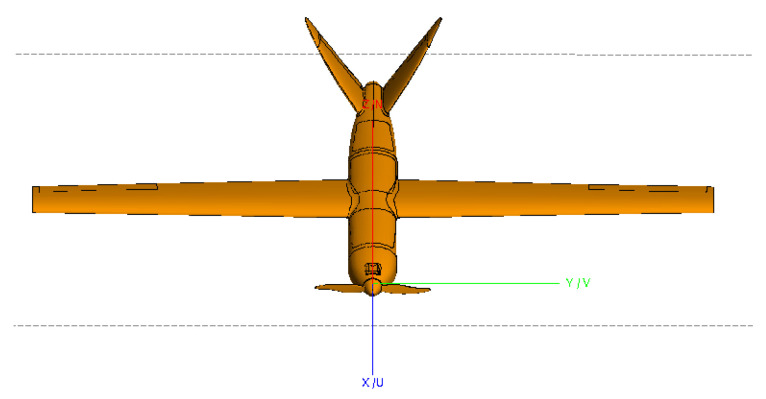
The CAD model of the aircraft.

**Figure 6 sensors-24-06315-f006:**
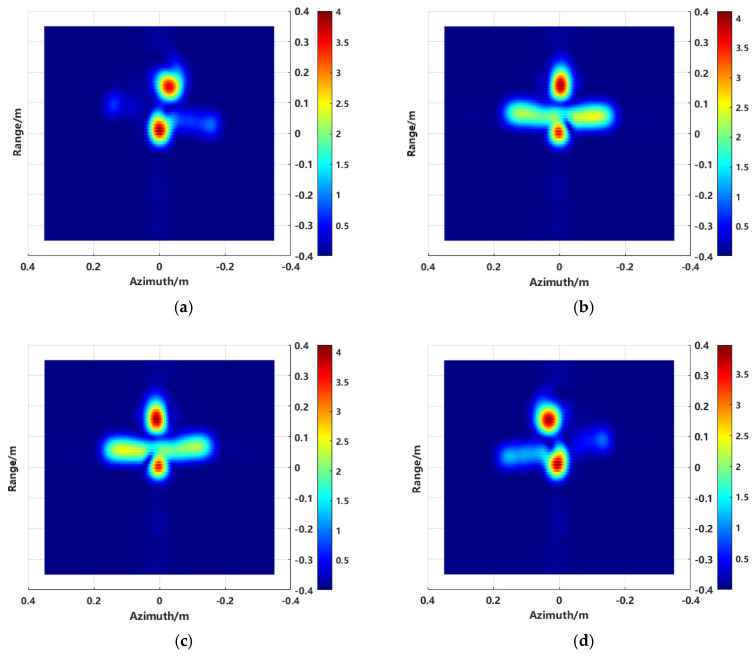
Imaging results with different sub-aperture center angles: (**a**) −12°, (**b**) −2°, (**c**) 2°, and (**d**) 10°.

**Figure 7 sensors-24-06315-f007:**
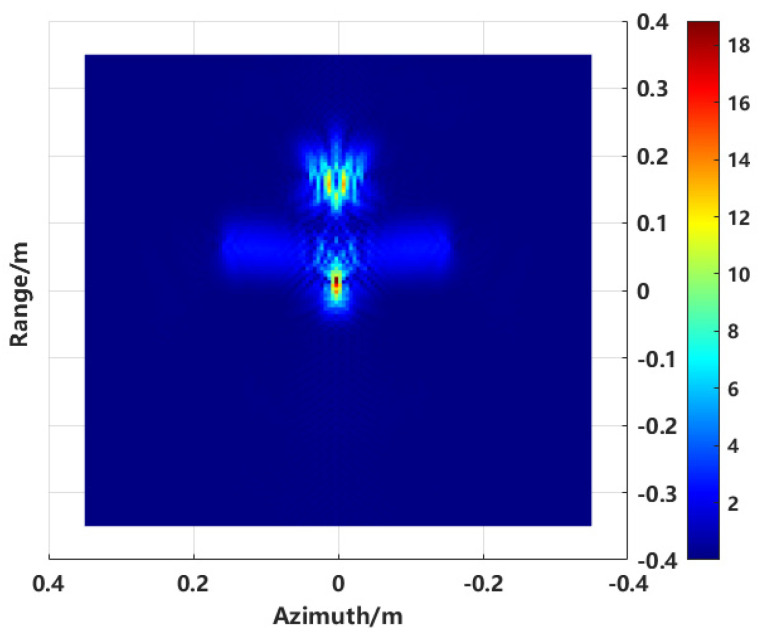
Enhanced imaging result.

**Figure 8 sensors-24-06315-f008:**
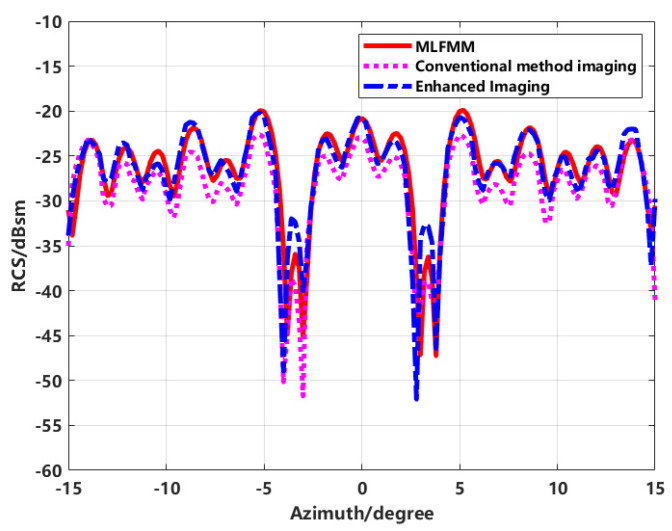
Comparison of RCS inverting results from two imaging methods with MLFMM calculation results.

**Figure 9 sensors-24-06315-f009:**
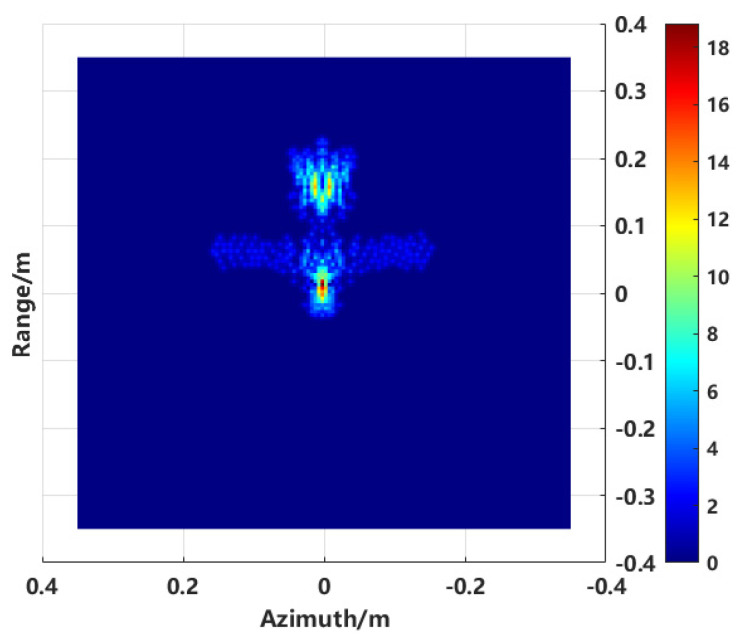
Scattering centers extraction combined enhanced imaging with sequence CLEAN.

**Figure 10 sensors-24-06315-f010:**
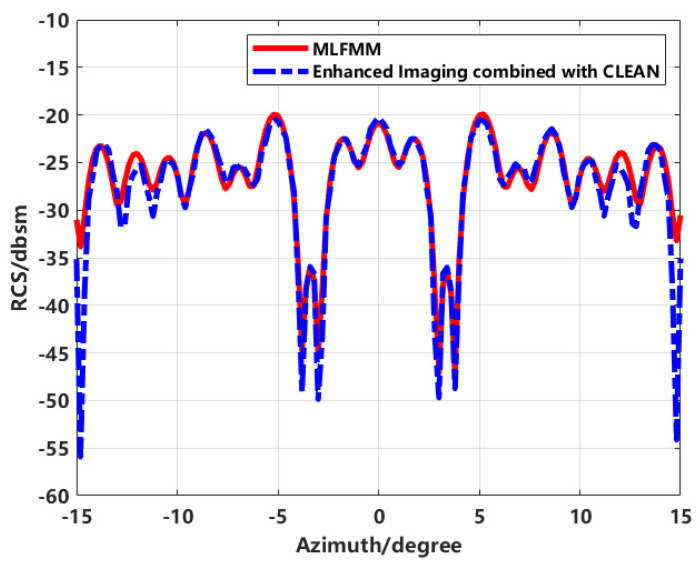
The comparison of inverting RCS and MLFMM calculations by the proposed method.

**Figure 11 sensors-24-06315-f011:**
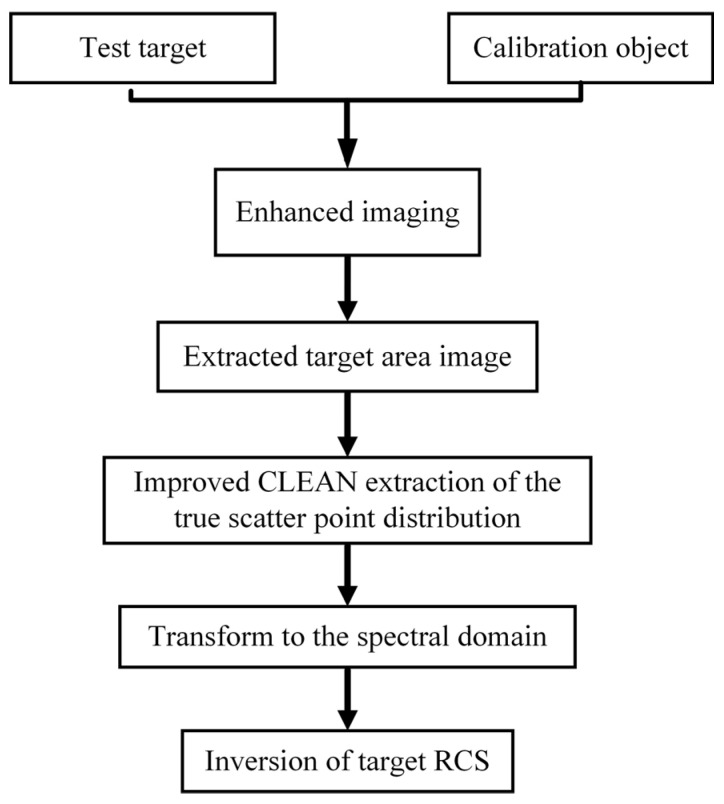
Overall workflow of imaging-based RCS measurement process.

**Figure 12 sensors-24-06315-f012:**
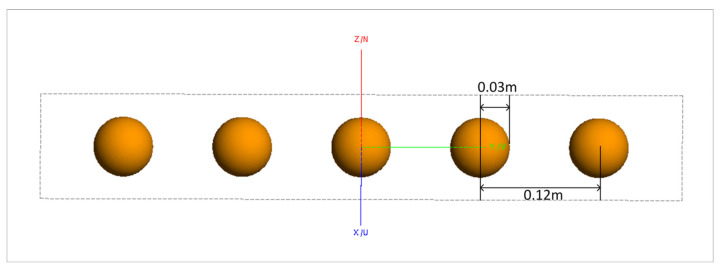
Simulation model of five metal spheres.

**Figure 13 sensors-24-06315-f013:**
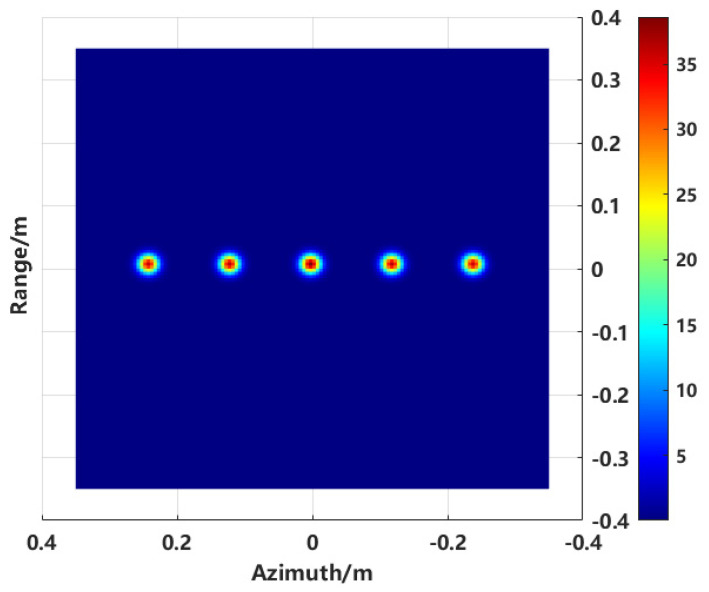
Distribution of scattering centers.

**Figure 14 sensors-24-06315-f014:**
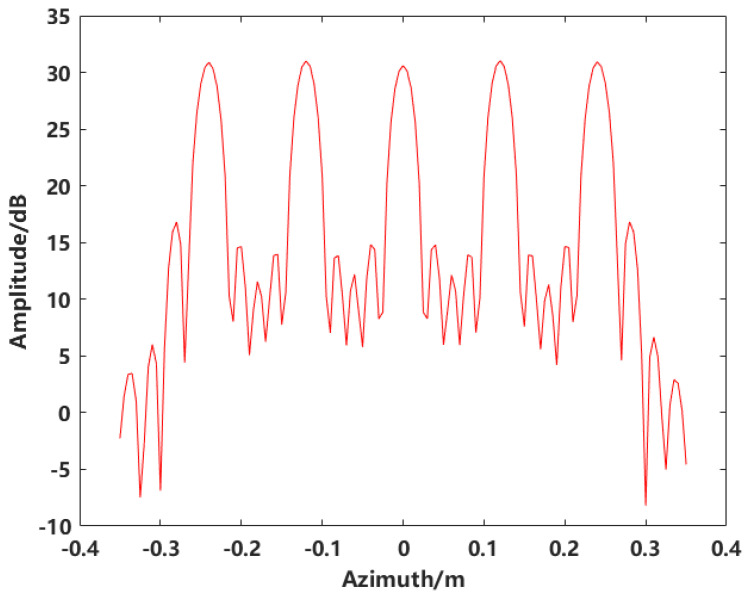
Azimuthal profile of conventional imaging.

**Figure 15 sensors-24-06315-f015:**
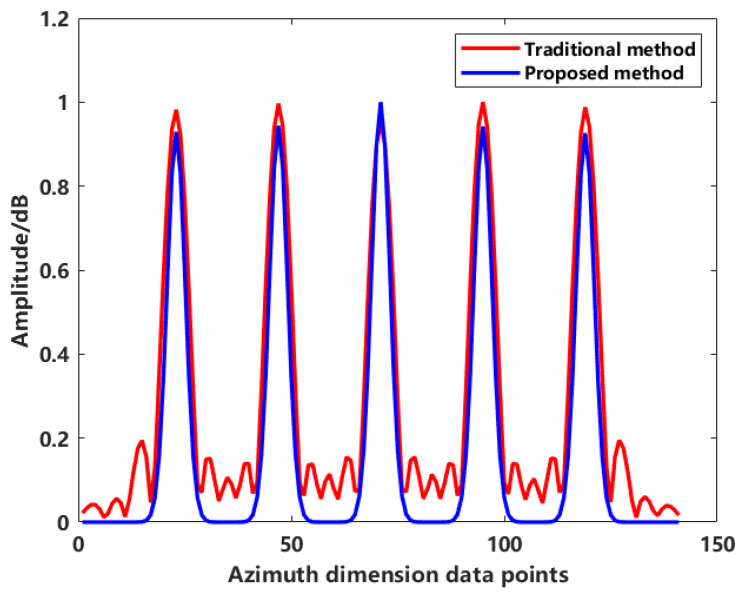
Normalized distribution of imaging profiles for conventional and proposed methods.

**Figure 16 sensors-24-06315-f016:**
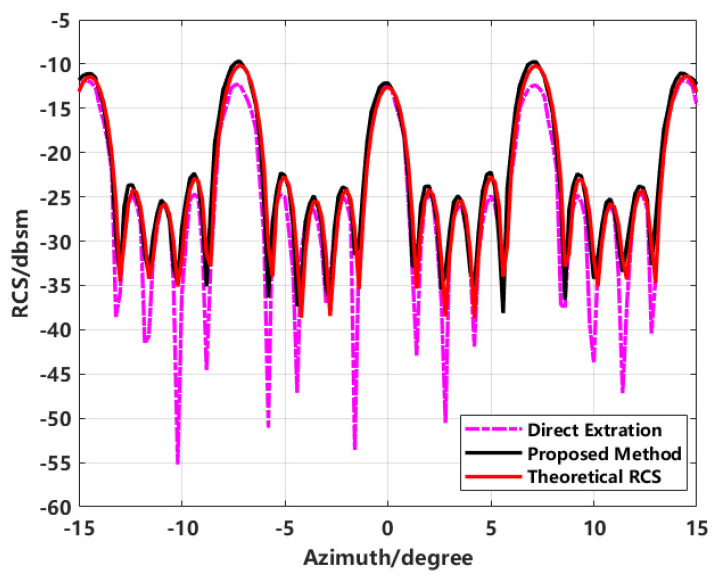
Comparison of inverted RCS from imaging with theoretical RCS.

**Figure 17 sensors-24-06315-f017:**
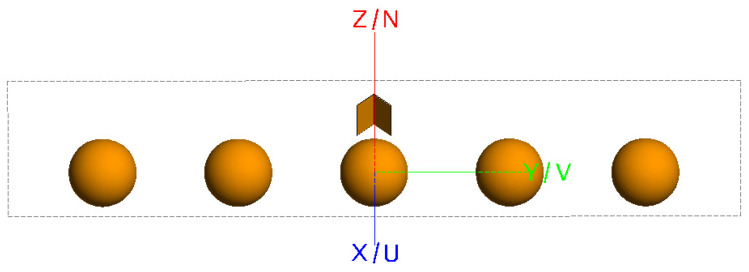
Model diagram with dihedral structure as interference source.

**Figure 18 sensors-24-06315-f018:**
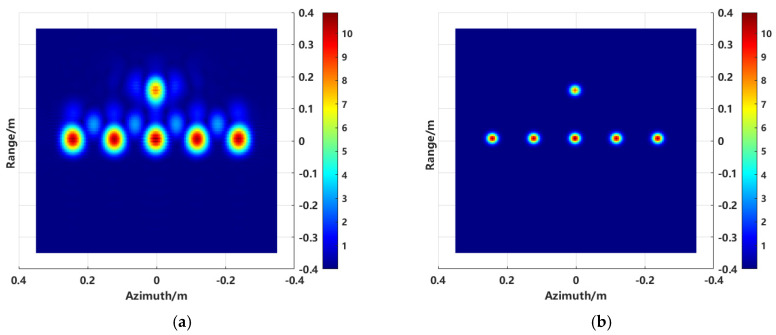
Imaging results. (**a**) Conventional imaging results, and (**b**) imaging results using the proposed method.

**Figure 19 sensors-24-06315-f019:**
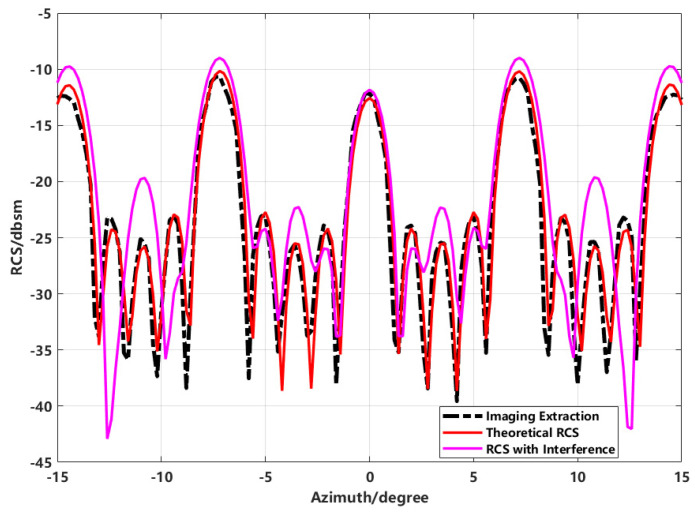
Comparison of results from various processing methods.

**Figure 20 sensors-24-06315-f020:**
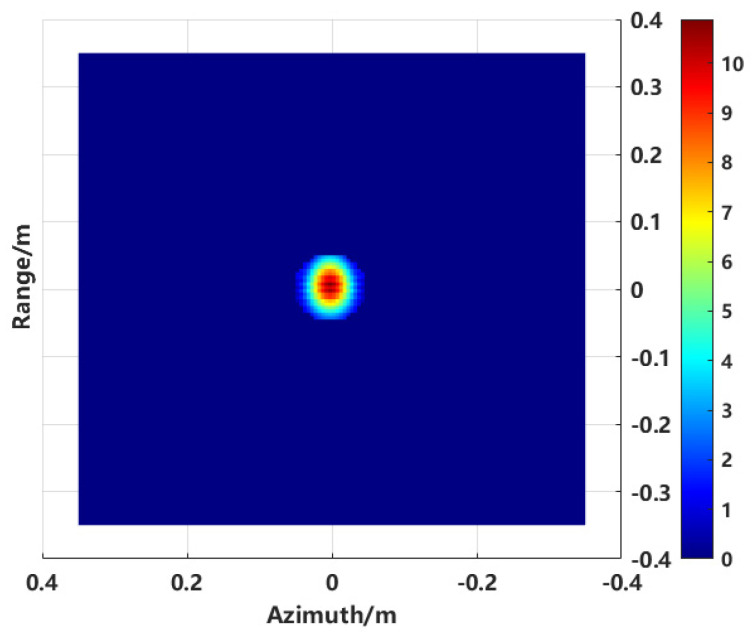
Extracted ROI image.

**Figure 21 sensors-24-06315-f021:**
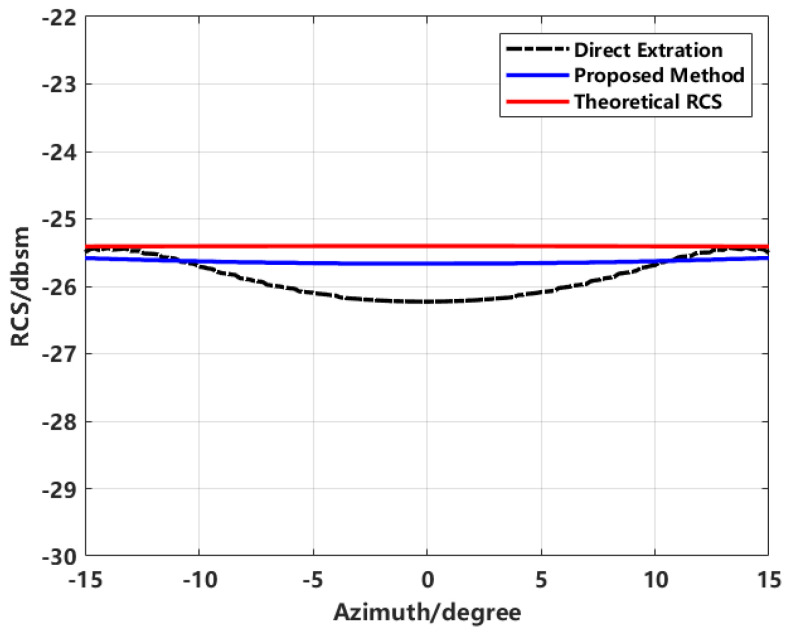
Comparison of local extraction.

**Figure 22 sensors-24-06315-f022:**
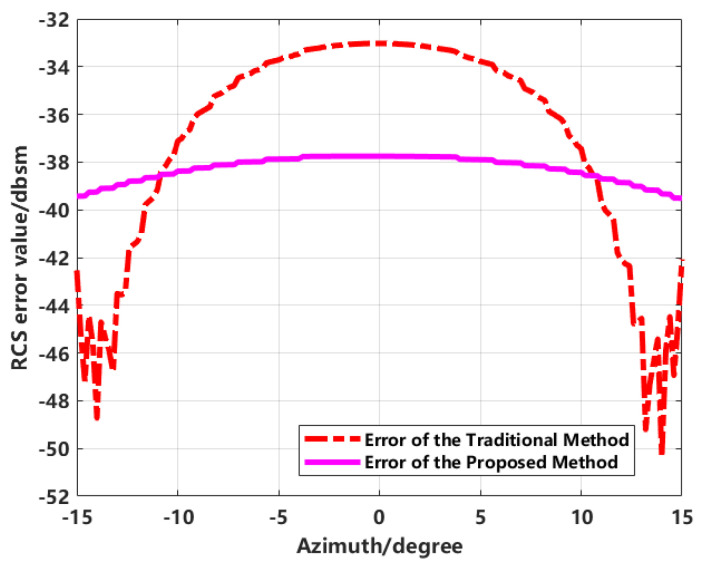
Error comparison.

**Figure 23 sensors-24-06315-f023:**
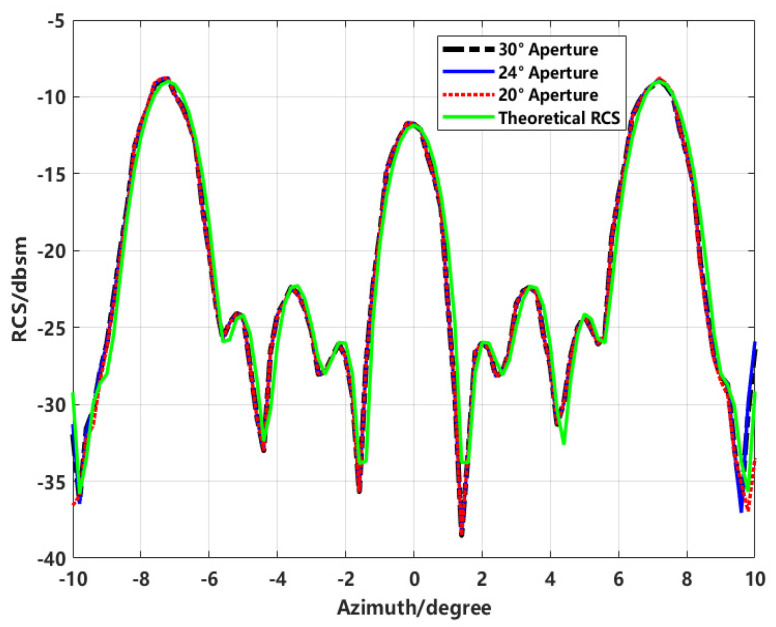
Comparison of inversion results with different aperture angles.

**Figure 24 sensors-24-06315-f024:**
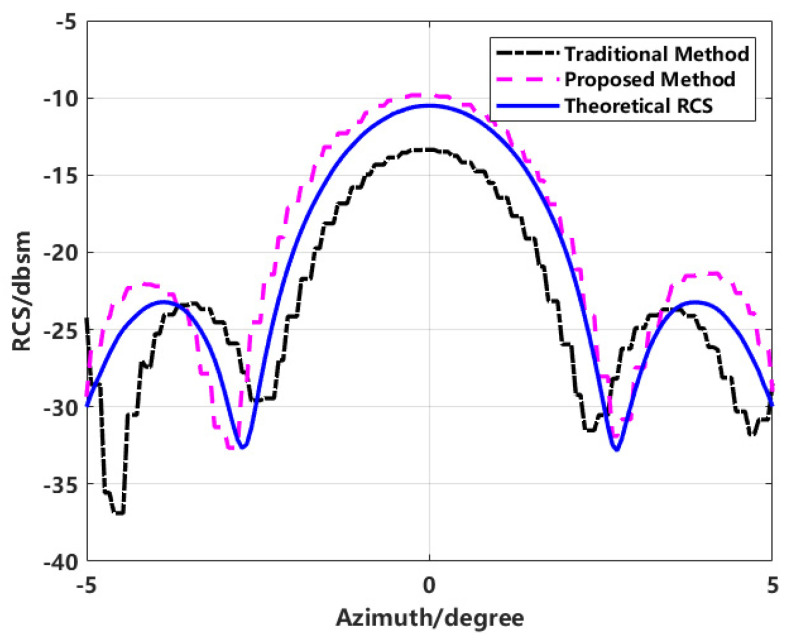
Comparison of RCS calculation results when resolution exceeds target spacing.

**Figure 25 sensors-24-06315-f025:**
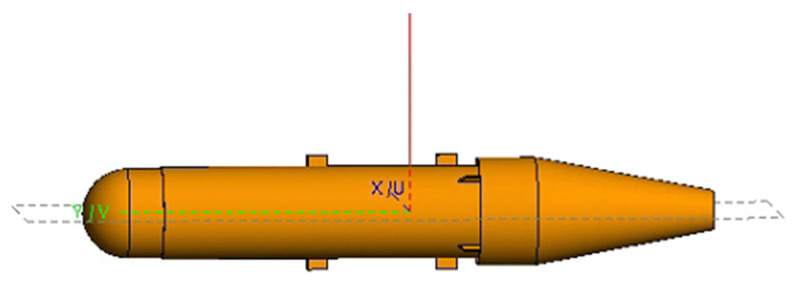
Model of thin and long circular pod.

**Figure 26 sensors-24-06315-f026:**
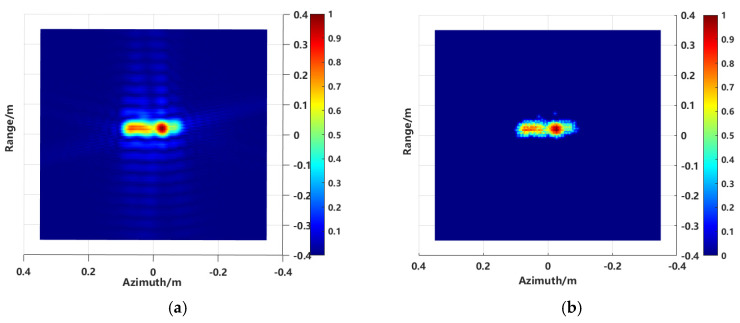
The 2D imaging results. (**a**) The conventional imaging result, and (**b**) the imaging result using the proposed method.

**Figure 27 sensors-24-06315-f027:**
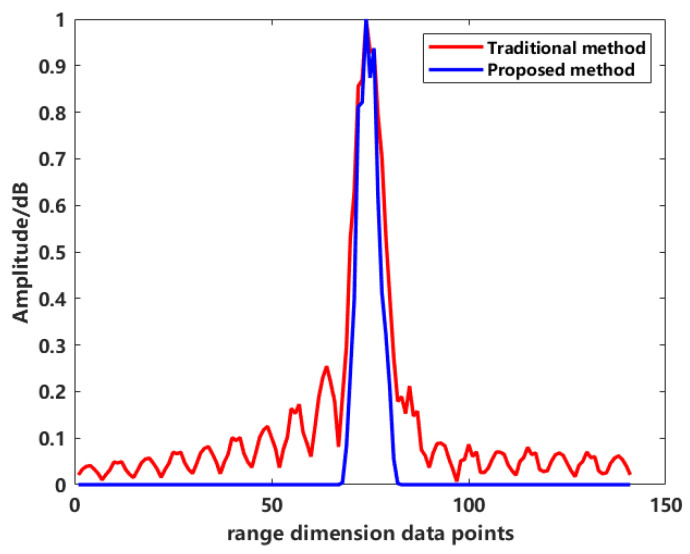
Distance profile of 2D imaging using different methods.

**Figure 28 sensors-24-06315-f028:**
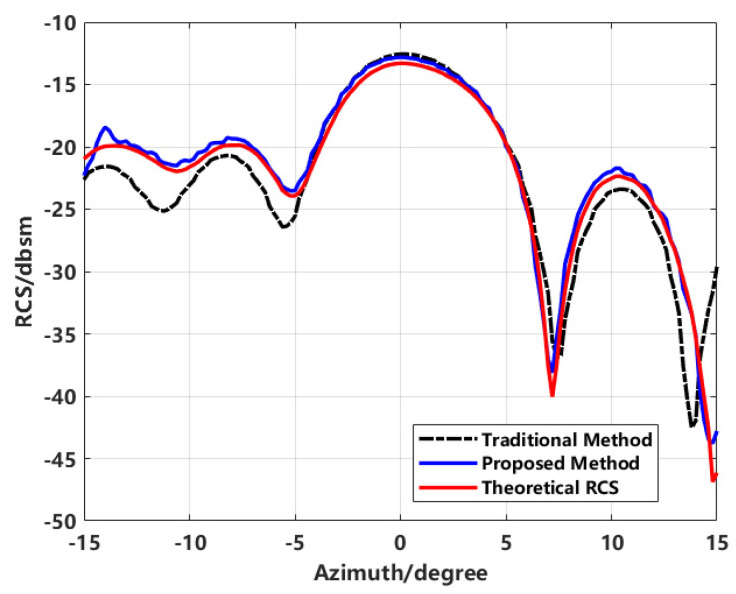
Comparison of results from various processing methods.

**Figure 29 sensors-24-06315-f029:**
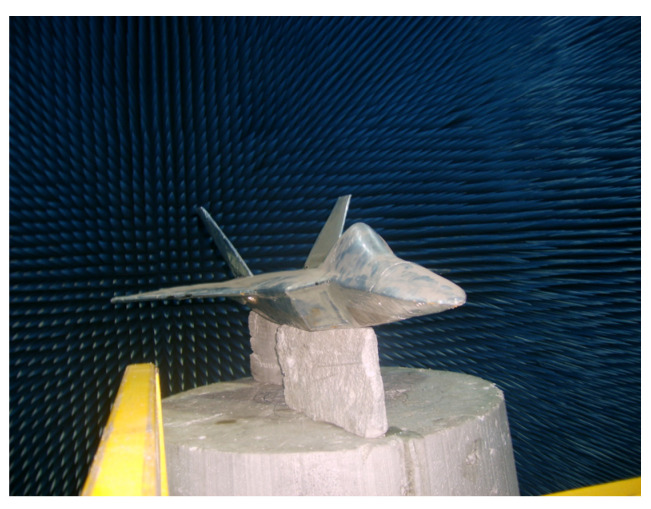
Optical image of model.

**Figure 30 sensors-24-06315-f030:**
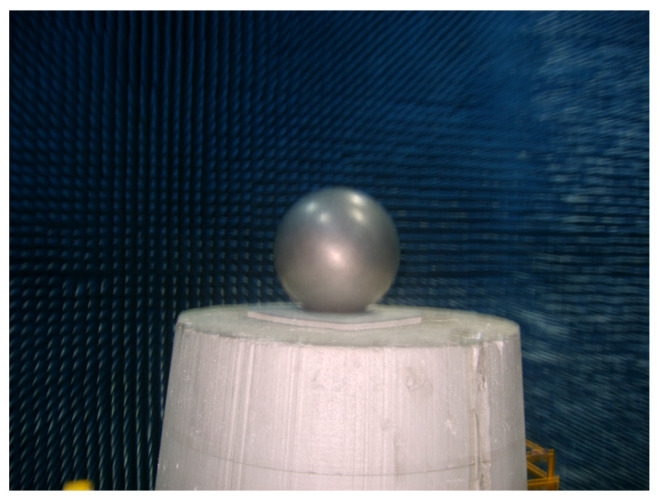
Calibration target.

**Figure 31 sensors-24-06315-f031:**
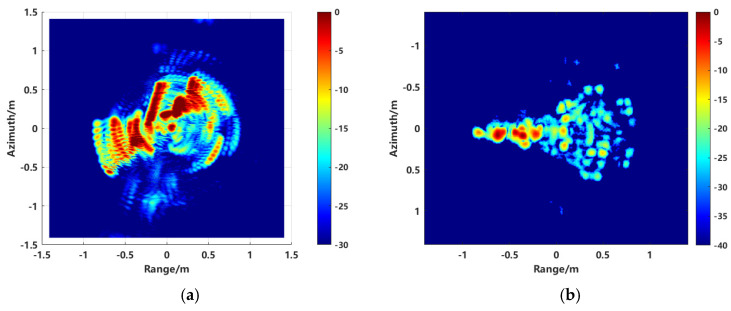
Imaging results. (**a**) Imaging offsets with different apertures, and (**b**) scattering center distribution.

**Figure 32 sensors-24-06315-f032:**
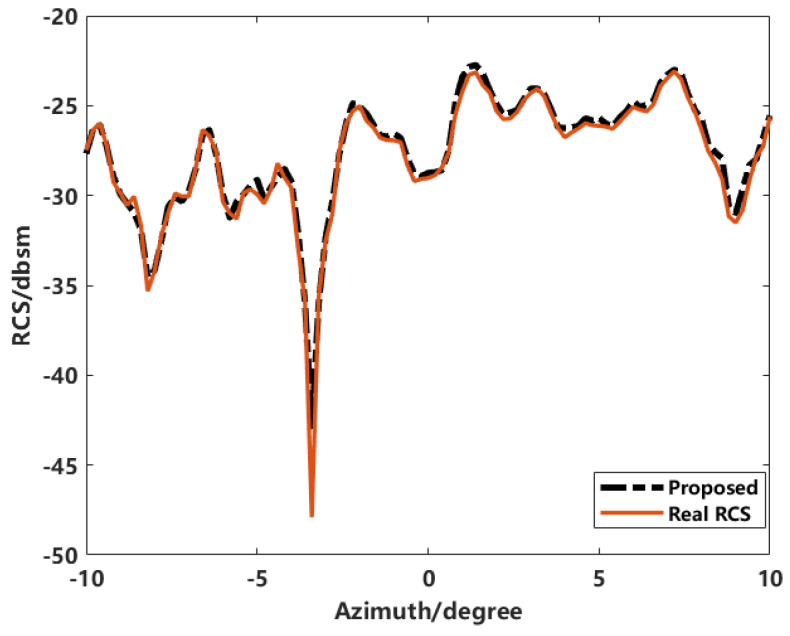
Comparison of results.

**Figure 33 sensors-24-06315-f033:**
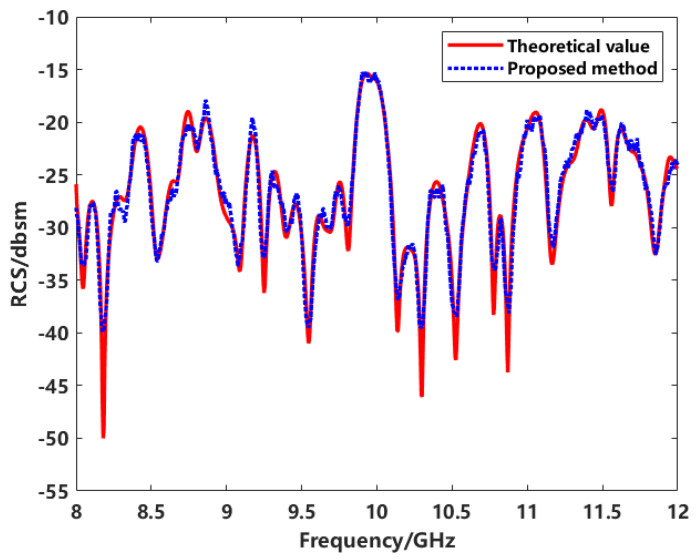
The frequency response of the RCS.

**Table 1 sensors-24-06315-t001:** Extraction position errors.

Target Object Name	Simulation Coordinate	Extract Coordinates	Error
Ball 1	(0, −0.24)	(−0.015, −0.235)	(0.015, 0.005)
Ball 2	(0, −0.12)	(−0.015, −0.115)	(0.015, 0.005)
Ball 3	(0, 0)	(−0.01, 0)	(0.01, 0)
Ball 4	(0, 0.12)	(−0.01, 0.12)	(0.01, 0)
Ball 5	(0, 0.24)	(−0.02, 0.24)	(0.02, 0)

**Table 2 sensors-24-06315-t002:** Comparison of Peak Sidelobe Ratio Values.

Method	Traditional Method	Proposed Method
SLR	−16.7027 dB	−21.0652 dB

**Table 3 sensors-24-06315-t003:** Inversion errors for different aperture sizes.

Aperture Size	MAE (dBsm)	RMSE (dBsm)
30°	1.2203	1.7497
24°	1.2347	1.7853
20°	1.2390	1.8250

**Table 4 sensors-24-06315-t004:** Comparison of Peak Sidelobe Ratio Values.

Method	Traditional Method	Proposed Method
SLR	−9.6141 dB	−12.582 dB

## Data Availability

The data presented in this study are available on request from the corresponding author after obtaining permission from an authorized person.

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
