# Peer review of "Radar Target Radar Cross-Section Measurement Based on Enhanced Imaging and Scattering Center Extraction"

_sensors, 2024, doi:10.3390/s24196315_

Round 1
Reviewer 1 Report
Comments and Suggestions for Authors
1. The proposed method should be compared in terms of performaces with other state-of-art techniques for RCS measurement, since the present manuscript is compared only with conventional technique (Back projection).
2. The appropriate definition of RCS Error must be provided. In the abstract, the value of RCS error is given with absolute value of 0.2 dBsm(nearly 1 m2), which means that low RCS under +10 dBsm(10 m2) cannot be measured with this algorithm.
3. Several figures are not explained sufficiently with the full information.
For example of Figure 7, information of frequency and angular samples is not given in the manuscript.
4. This manucript proposed the sequence CLEAN technique for reducing the sidelobe level. Then, quantitave comaparison is required, so that the reader can readily understand that the RCS can be affected by the extent of sidelobe.
Author Response
The authors would like to thank the reviewer for giving the comments on improving the quality of the paper. Thank you very much. We would like to carefully clarify as follows:
Q1: The proposed method should be compared in terms of performances with other state-of-art techniques for RCS measurement, since the present manuscript is compared only with conventional technique (Back projection).
Response 1:
The primary method for analyzing the scattering characteristics of a target and evaluating its Radar Cross Section (RCS) is through direct RCS measurement. However, as measurement precision requirements have increased—particularly with the need to mitigate environmental interference—RCS measurement techniques based on imaging have been developed. These techniques offer the dual advantages of visual diagnosis for defects in the target's scattering characteristics and precise extraction of localized scattering properties of interest. This paper focuses on the RCS measurement method based on imaging and introduces improvements to the current imaging-based RCS measurement techniques, significantly enhancing their performance. In the performance comparison experiments, we compare our proposed method with both direct RCS measurement techniques and traditional imaging-based RCS extraction methods to demonstrate the superiority of our approach.
Furthermore, imaging-based RCS measurement methods have gained widespread attention due to their interference resistance capabilities and have led to several research developments in this area. Most of these methods use the BP algorithm for imaging, as evidenced by the studies in references [1-3], which utilize BP imaging algorithms for RCS measurement. This paper builds on this foundation, considering that the BP algorithm can precisely compute the slant range from each scattering point to the radar and perform coherent integration in the time domain along the scattering point trajectory, thereby achieving higher resolution even at large angles. Compared to other traditional imaging methods, the BP algorithm avoids the need for cross-resolving unit motion correction, making it widely adopted. Therefore, this paper includes commonly used BP-based imaging methods in the performance comparison.
[1] Larsson C. Nearfield RCS measurements of full scale targets using ISAR[C]. 36th Annual Symposium of the Antenna Measurements Techniques Association. Antenna Measurement Techniques Association, 2014: 79-84.
[2] J. Dang; Y. Luo; C. Hu. A Local RCS Diagnosis Method Based on Near Field Measurement. IEEE Transactions on Instrumentation and Measurement, 2023, 72, 1-9.
[3] Qunting Ren, Xiao Wei, Chao Gao, Ming Lyu. Three-dimensional BP Imaging Algorithm using MIMO System, Procedia Computer Science. 2021, 187, 103-108.
Q2: The appropriate definition of RCS Error must be provided. In the abstract, the value of RCS error is given with absolute value of 0.2 dBsm(nearly 1 m2), which means that low RCS under +10 dBsm(10 m2) cannot be measured with this algorithm.
Response 2:
We agree with your feedback that the definition of error in the manuscript is inaccurate and could lead to reader misunderstanding. In this paper, the RCS error refers solely to the relative error between the theoretical RCS of the target and the RCS measured using our method (in units of dBsm), rather than an absolute "error magnitude" expressed in dBsm. When converting the relative error in this paper to units of (m²), the 0.2 dBsm error value mentioned in the abstract becomes ambiguous. Therefore, we will remove the error description from the abstract and provide the RCS error definition in the main text: the error is calculated as the numerical difference between the RCS obtained using our algorithm and the reference value, measured in m², and then converted using , with the result expressed in dBsm. For example, if the actual error is 0.01 m², it converts to -20 dBsm. Figure 22 in the main text will be redrawn accordingly, and the analysis text below Figure 22 will be revised to include the updated RCS error definition and modifications.
The modifications are as follows:
Figure 22. Error comparison results.
According to Figure 22, for simple geometric targets, the RCS error using both the direct extraction method and the proposed method ranges from -48 dBsm to -33 dBsm. Here, the RCS error is defined as the difference between the RCS obtained using the proposed method and the reference value, initially measured in m², and then converted using , with the result expressed in dBsm. However, the direct extraction method includes sidelobes within a certain range around the scattering points when extracting the image, leading to errors fluctuating between -48 dBsm and -33 dBsm compared to the MLFMM simulation results. The enhanced imaging combined with the sequence CLEAN technique effectively suppresses sidelobe interference from surrounding targets on the central sphere, keeping the inversion error consistently within approximately -39 dBsm across various azimuths. Overall, the proposed method significantly improves the accuracy of RCS measurement.
Q3: Several figures are not explained sufficiently with the full information.
For example of Figure 7, information of frequency and angular samples is not given in the manuscript.
Response 3:
Thank you for your valuable feedback. We have reviewed all the figures in the manuscript and added detailed descriptions where needed. The added descriptions are as follows:
The following text has been added above Figure 7: Imaging of the model in Figure 5 using the proposed method with an aperture size ranging from -30° to 30° and a frequency range of 8 to 12 GHz. The sub-aperture synthesis was performed using the proposed method, and the imaging result is shown in Figure 7.
The text above Figure 9 has been revised to: A new two-dimensional image is generated by applying the scattering center extraction method to the image shown in Figure 7, as presented in Figure 9.
Q4: This manuscript proposed the sequence CLEAN technique for reducing the sidelobe level. Then, quantitative comparison is required, so that the reader can readily understand that the RCS can be affected by the extent of sidelobe.
Response 4:
In the ISAR imaging model, the ideal point target is represented by a two-dimensional SINC function for its backscattering center. Therefore, ISAR imaging is affected by the SINC function’s sidelobe superposition issue. To address this, a point spread function (PSF) containing only the main lobe is used to construct the scattering points of the target. In the improved CLEAN technique proposed in this paper, a two-dimensional Gaussian function is chosen as the PSF. By iteratively finding the sequence that minimizes the reduction in image energy, the technique reconstructs the true scattering points to ensure the accuracy of scattering center extraction.
We imaged the target model in Figure 12 using both the traditional method and the sequence CLEAN method. The scattering points from the central horizontal cross-section were extracted and normalized. The resulting two-dimensional horizontal cross-sectional images, shown in the figure, illustrate the distribution of main lobes and sidelobes for both methods. It is evident that the proposed method significantly suppresses the sidelobes between scattering points. Compared to the traditional method, the main lobe remains unchanged in width, while the sidelobes decay much more rapidly.
Figure 15. Normalized Distribution of Imaging Profiles for Conventional and Proposed Methods
The Integration Side Lobe Ratio (ISLR) refers to the ratio of the energy outside the impulse response resolution cell to the energy within the resolution cell. Equation 13 provides the formula for calculating the ISLR.
(13)
Where, represents the energy within the main lobe, while represents the energy within the sidelobes.
A lower ISLR in SAR images indicates that the dark areas in the image are less likely to be affected by adjacent strong scatterers. This paper uses the Integration Side Lobe Ratio (ISLR) to quantitatively assess the advantages of the proposed method over traditional methods in reducing sidelobes. The quantitative results are presented in Table 2.
Table 2. Comparison of Peak Side Lobe Ratio Values
Method |
Traditional Method |
Proposed Method |
|
SLR |
-16.7027dB |
-21.0652dB |
|
In the inversion results shown in Figure 15, the comparison between the results obtained using the proposed method and those obtained through direct imaging extraction demonstrates that the proposed method not only effectively suppresses sidelobes but also significantly improves the RCS measurement results.

Reviewer 2 Report
Comments and Suggestions for Authors
This paper proposes a target RCS measurement method based on enhanced imaging and scattering center extraction with the sequence CLEAN algorithm. Experimental results demonstrate that this method achieves higher precision in RCS measurement of complex targets. There are some issues need to be considered.
1. For the proposed method, the quality of the imaging directly impacts the measurement accuracy of the target RCS. How much does the imaging resolution affect the measurement accuracy of RCS? It is recommended to give an analysis under different imaging resolution.
2. As the equivalent scattering centers of the target vary with different viewing angles, known as the anisotropic characteristics, the sub-aperture synthesis and image fusion by rotation and alignment of sub-images will obtain the average imaging. How does such an average imaging ensure the accuracy of the true scattering characteristics? What does the scattering centers distribution shown in Fig. 7 correspond to?
3. The proposed RCS measurement method based on the imaging of target requires wideband signals. The RCS measurement is obtained with respect to azimuthal angles. Is it possible to obtain the RCS measurement at different frequencies, i.e. the frequency response of the RCS?
Comments on the Quality of English LanguageThe quality of English language is fine.
Author Response
The authors would like to thank the reviewer for giving the comments on improving the quality of the paper. Thank you very much.
The detailed answers to corresponding questions are as follow.
Q1: For the proposed method, the quality of the imaging directly impacts the measurement accuracy of the target RCS. How much does the imaging resolution affect the measurement accuracy of RCS? It is recommended to give an analysis under different imaging resolution.
Response1:
Theoretically, resolution affects methods for "imaging-based RCS measurement." To further illustrate this impact, we use simulations. For a model with five spheres arranged linearly, where the distance between adjacent targets is 0.08 meters, and the simulation is set with an azimuthal resolution of 0.1011 meters, the scattering points of the targets are not distinguishable. RCS calculations were performed using both traditional methods and the proposed method, and the final results are compared with the theoretical RCS curve as shown in the figure 24.
Figure 24. Comparison of RCS Calculation Results When Resolution Exceeds Target Spacing
The results show that when the resolution is greater than the minimum resolvable distance between targets, the proposed method provides a better match with the data curve compared to traditional methods. However, some corresponding errors are inevitably present. This indicates that, regardless of whether the resolution meets the resolvability criteria, the proposed method maximizes the integration of all scattering information from the targets, offering higher measurement accuracy.
In the main text, we added the following content regarding the impact of resolution:
It is well known that the azimuthal resolution is related to the size of the synthetic aperture and the center frequency. In the above experiment, using the model shown in Figure 16 with a distance of 0.12 meters between the centers of two spheres, the azimuthal resolution in experiments with different aperture angles was always less than the distance between the targets, meaning that any two targets could always be resolved. Thus, the proposed method achieves high measurement accuracy for RCS when targets are resolvable. To verify the effectiveness of the proposed method when targets are not resolvable, we simulated the model in Figure 16 with the distance between the centers of the two spheres set to 0.08 meters. With an azimuthal aperture angle of 10° and a test frequency band of 8-9 GHz, the azimuthal resolution calculates to 0.1011 meters, and the range resolution calculates to 0.15 meters. The comparison of the RCS from the imaging calculation model with the theoretical RCS is shown in Figure 24.
Figure 24. Comparison of RCS Calculation Results When Resolution Exceeds Target Spacing
The results show that when the resolution exceeds the minimum resolvable distance between targets, the proposed method offers a better fit with the data curve compared to traditional methods. This indicates that, regardless of whether the resolution meets the resolvability criteria, the proposed method integrates all scattering information from the targets to the greatest extent possible, thereby providing higher measurement accuracy.
Q2:As the equivalent scattering centers of the target vary with different viewing angles, known as the anisotropic characteristics, the sub-aperture synthesis and image fusion by rotation and alignment of sub-images will obtain the average imaging. How does such an average imaging ensure the accuracy of the true scattering characteristics? What does the scattering centers distribution shown in Fig. 7 correspond to?
Response 2:
After using the sub-aperture method, the scattering characteristics of the scattering points vary continuously in the azimuthal direction rather than abruptly, which better aligns with the scattering characteristics of the target from different angles. By integrating multi-view information, each view provides scattering information from a different observation angle, complementing each other. This allows for a comprehensive consideration of variations in the position and intensity of scattering centers from different angles. When these sub-image information are linearly averaged, the overall scattering information is more comprehensively reflected. Linear averaging of multiple sub-apertures can suppress sidelobe effects caused by insufficient or incomplete sampling of individual sub-apertures. This processing ensures that high-quality data contribute more to the final image, reducing the impact of noise and spurious scatterers.
In image registration and fusion methods, a logical filtering fusion method is also available. However, this method only compares the pixel values of each sub-image and selects the maximum value to enhance the image. For targets with different composite structures, the scattering characteristics in different component areas may not be distinguishable under a single sub-aperture, and weak scattering areas will be neglected. As a result, the overall scattering characteristics of the target will only reflect strong scattering regions. The average weighting approach, to some extent, acts as a "smoothing" mechanism, reducing errors caused by sidelobes. Additionally, through multiple sampling integrations, the real scattering center's position and intensity are reinforced while anisotropy-induced deviations are minimized. This approach helps obtain the global scattering characteristics of the target rather than those from a local or single-view perspective, thereby enhancing the accuracy of the final imaging.
The scattering center distribution in Figure 7 corresponds to the aircraft model in Figure 5. This scattering center distribution is solely based on the imaging results formed by sub-aperture integration and image fusion enhancement methods.
Q3:The proposed RCS measurement method based on the imaging of target requires wideband signals. The RCS measurement is obtained with respect to azimuthal angles. Is it possible to obtain the RCS measurement at different frequencies, i.e. the frequency response of the RCS?
Response 3:
Obtaining RCS measurements at different frequencies is possible by keeping the azimuthal angle fixed, thus deriving the frequency response of the RCS. I will include this experimental result in the main text. The additional content is as follows:
Figure 29. The frequency response of the RCS.
Figure 29 shows the frequency response curve of the RCS with a test frequency range of 8-12 GHz. According to the experimental results, the RCS measurement curves obtained using the proposed method exhibit good agreement with the true RCS curves at various frequency points. This indicates that the method demonstrates practical applicability.
Round 2
Reviewer 1 Report
Comments and Suggestions for Authors
The authors addressed all questions, suggestions, and comments of this Reviewer very carefully, clearly, and thoroughly. They revised the manuscript very well according to Reviewer' Comments.
The revised manuscript is in good shape, however, this Reviewer has one suggestion.
The proposed method is only demonstrated with point-source targets in Section 3 for enhancing image, reducing the side lobe level, and analyzing the error level. Another demonstration example can be suggested for a thin and long circular pod with length of 0.5 m.
Author Response
The authors would like to thank the reviewer for giving the comments on improving the quality of the paper. Thank you very much.
Q1: The authors addressed all questions, suggestions, and comments of this Reviewer very carefully, clearly, and thoroughly. They revised the manuscript very well according to Reviewer' Comments.
The revised manuscript is in good shape, however, this Reviewer has one suggestion.
The proposed method is only demonstrated with point-source targets in Section 3 for enhancing image, reducing the side lobe level, and analysing the error level. Another demonstration example can be suggested for a thin and long circular pod with length of 0.5 m.
Responses :
We have incorporated the model test validation as per your suggestion. The model diagram is shown in Figure 25. You recommended a model with the length of 0.5m, unfortunately, due to limited computational resources (as the revisions requested to be submitted within two days), the calculation time for a 0.5m model is quite slow, with the mesh division reaching several hundred thousand elements, and the computation is still in progress.
However, we have completed the same model calculation with a reduced length of 0.2m, which improved the computational efficiency. The validation of the method proposed in the paper was conducted using this 0.2m model.
Figure25. Model of a thin and long circular pod
The specific simulation parameters are as follows: a center frequency of 10 GHz, a bandwidth of 4 GHz, with 101 sampled frequency points. The azimuth angle range is -15° to 15°, with 151 sampling points. Figure 1 shows the simulation target, which is a sprayer pod. The total length of the target is 0.2 meters.
The verification was carried out based on the given target model. The imaging results of both the traditional imaging method and the proposed method are shown in Figures 2(a) and (b), respectively.
(a) (b)
Figure 26. 2D imaging results using two methods. (a) Conventional imaging result, and (b) imaging result using the proposed method.
From the visual analysis of the 2D imaging results, the traditional imaging method exhibits speckle noise in the range direction, whereas the proposed method effectively filters out clutter outside of the target, resulting in a cleaner background level.
To further analyze the sidelobes in the 2D images more intuitively, the central range profile was extracted for both imaging methods. The range profile of the traditional method and the scattering center extraction method's central range profile is shown in Figure 27.
Figure27. Distance profile of 2D imaging using different methods
Figure 27 shows the normalized results. In the range profile obtained using the traditional method, the central point corresponds to the target area, and the sidelobes on both sides of the main lobe exhibit oscillations characteristic of a SINC function. These oscillations indicate that the 2D image generated by the traditional method is affected by significant clutter. However, analyzing the results of the proposed method in Figure 27, the central position corresponds to the target area, and the middle section represents the scattering points that correspond to the target’s physical structure. Outside the target area, the sidelobes in the profile are clearly visible. From the vertical axis of the graph, it is evident that the sidelobe values drop sharply compared to the traditional method, effectively suppressing clutter.
To quantitatively assess the sidelobe suppression effect, we also used the integrated sidelobe ratio (ISLR) as an evaluation metric. A lower ISLR indicates that dark areas in the image are less affected by nearby strong scatterers. The quantitative results are shown in Table 4.
Table 4. Comparison of Peak Side Lobe Ratio Values
Method |
Traditional Method |
Proposed Method |
|
SLR |
-9.6141dB |
-12.5862dB |
|
The analysis above demonstrates the advantages of the proposed method in sidelobe suppression, which serves as the basis for comparing the final RCS measurement accuracy. The final RCS measurement accuracy is calculated using both the traditional method and the enhanced imaging scattering center method proposed in this study. The results are then compared with the theoretical RCS values, as shown in Figure 28.
Figure 28. Comparison of the results from various processing methods.
The black dashed line and the blue solid line represent the RCS curves obtained from the traditional method and the proposed method, respectively. In the azimuth range of -15° to -5°, the RCS calculated by the traditional method shows a significant relative error compared to the theoretical value. This discrepancy primarily arises from the tip effect of the cone and the mutual coupling of various small components, which introduce clutter and false points in the two-dimensional image. During the scattering inversion process, these false points and clutter negatively impact the RCS calculation. In contrast, the curve generated by the proposed method aligns closely with the theoretical values, demonstrating high-quality measurement accuracy.

Reviewer 2 Report
Comments and Suggestions for Authors
The previous concerns have been addressed. New simulation results are added, which makes the paper clearer.
Comments on the Quality of English LanguageThe quality of English language is fine.
Author Response
The authors would like to thank the reviewer for giving the comments on improving the quality of the paper. Thank you very much.